# SUMOylation Modulates Reactive Oxygen Species (ROS) Levels and Acts as a Protective Mechanism in the Type 2 Model of Diabetic Peripheral Neuropathy

**DOI:** 10.3390/cells12212511

**Published:** 2023-10-24

**Authors:** Nicolas Mandel, Michael Büttner, Gernot Poschet, Rohini Kuner, Nitin Agarwal

**Affiliations:** 1Institute of Pharmacology, Medical Faculty Heidelberg, Heidelberg University, Im Neuenheimer Feld 366, 69120 Heidelberg, Germanyrohini.kuner@pharma.uni-heidelberg.de (R.K.); 2Centre for Organismal Studies (COS), University of Heidelberg, Im Neuenheimer Feld 360, 69120 Heidelberg, Germany

**Keywords:** hyperglycaemia, high-fat diet, SUMOylation, reactive oxygen species, malate dehydrogenase 2, respiratory chain

## Abstract

Diabetic peripheral neuropathy (DPN) is the prevalent type of peripheral neuropathy; it primarily impacts extremity nerves. Its multifaceted nature makes the molecular mechanisms of diabetic neuropathy intricate and incompletely elucidated. Several types of post-translational modifications (PTMs) have been implicated in the development and progression of DPN, including phosphorylation, glycation, acetylation and SUMOylation. SUMOylation involves the covalent attachment of small ubiquitin-like modifier (SUMO) proteins to target proteins, and it plays a role in various cellular processes, including protein localization, stability, and function. While the specific relationship between high blood glucose and SUMOylation is not extensively studied, recent evidence implies its involvement in the development of DPN in type 1 diabetes. In this study, we investigated the impact of SUMOylation on the onset and progression of DPN in a type 2 diabetes model using genetically modified mutant mice lacking SUMOylation, specifically in peripheral sensory neurons (SNS-Ubc9^−/−^). Behavioural measurement for evoked pain, morphological analyses of nerve fibre loss in the epidermis, measurement of reactive oxygen species (ROS) levels, and antioxidant molecules were analysed over several months in SUMOylation-deficient and control mice. Our longitudinal analysis at 30 weeks post-high-fat diet revealed that SNS-Ubc9^−/−^ mice exhibited earlier and more pronounced thermal and mechanical sensation loss and accelerated intraepidermal nerve fibre loss compared to control mice. Mechanistically, these changes are associated with increased levels of ROS both in sensory neuronal soma and in peripheral axonal nerve endings in SNS-Ubc9^−/−^ mice. In addition, we observed compromised detoxifying potential, impaired respiratory chain complexes, and reduced levels of protective lipids in sensory neurons upon deletion of SUMOylation in diabetic mice. Importantly, we also identified mitochondrial malate dehydrogenase (MDH2) as a SUMOylation target, the activity of which is negatively regulated by SUMOylation. Our results indicate that SUMOylation is an essential neuroprotective mechanism in sensory neurons in type 2 diabetes, the deletion of which causes oxidative stress and an impaired respiratory chain, resulting in energy depletion and subsequent damage to sensory neurons.

## 1. Introduction

Diabetes is a chronic medical condition that requires ongoing management. It is associated with significant healthcare costs, which are expected to rise further as the prevalence of diabetes increases [1]. In the United States alone, the total estimated cost of diabetes was USD 327 billion in 2017, which includes direct medical costs and indirect costs, such as lost productivity [2]. The high cost of diabetes management is due in part to the fact that it requires ongoing monitoring and treatment, including regular blood glucose testing, medications, and medical appointments. Diabetes affects the body’s ability to metabolize sugars, which is caused by either insufficient insulin production or resistance to insulin. There are two main types of diabetes: type 1 and type 2 diabetes. Type 1 is an autoimmune disease, where immune cells develop sensitivity against insulin-producing beta cells.

Type 2 diabetes is the most prevalent form; it usually occurs in adulthood, and it is related to obesity and a sedentary lifestyle. It results from insulin resistance causing a burden on pancreatic beta cells, which gradually lose their ability to produce insulin. If left untreated, both types 1 and 2 diabetes can lead to serious complications, such as nerve damage (diabetic neuropathy), kidney damage (diabetic nephropathy), and cardiovascular disease [3,4,5]. DPN, a common complication of diabetes, is peripheral nerve damage due to prolonged hyperglycaemia [3,6,7,8]. It affects various body parts, including the feet, legs, hands, arms, and internal organs, causing loss of sensation, numbness, tingling, and burning pain [9]. Around 50% of diabetic patients develop DPN incidentally [10]. Although the exact cause of its onset and progression is not well known, it is believed that hyperglycaemia damages blood vessels or disrupts neuronal metabolism, leading to toxic metabolite production and neuronal dysfunction [11]. Post-translational modifications (PTMs) of neuronal proteins, such as the formation of advanced glycation end-products, can contribute to oxidative stress and inflammation, ultimately leading to nerve damage [12,13,14]. SUMOylation, a critical post-translational modification (PTM), intricately orchestrates cellular functions by modulating individual protein activities and diverse pathways. Its crucial role in cell survival is underscored, particularly in neuroprotection against stroke [15], Alzheimer’s disease [16], and ischemic events [17], highlighting its significance in physiological processes. Recent studies suggest that SUMOylation may also play a role in the pathogenesis of diabetes [18,19]. While we have demonstrated SUMOylation’s protective role against diabetic neuropathy in type 1 diabetes, its relevance in type 2 diabetes, a prevalent disorder, remains unexplored.

SUMOylation is a process in which a small ubiquitin-like modifier (SUMO) is covalently attached to the lysine residue of a target protein. This modification regulates cellular processes, such as transcription, DNA repair, and protein localization [20,21]. A series of enzymatic reactions ensure the correct protein is targeted at the right time. The SUMO family includes three major paralogs: SUMO-1, SUMO-2, and SUMO-3. SUMO proteins are translated as inactive precursors, which are cleaved by SUMO-specific protease (SENP) enzymes to expose the C-terminal diglycine motif for conjugation. Activated by SAE1/2, SUMO is transferred to Ubc9, which binds to specific target proteins. E3 ligases and UBC9 transfer SUMO to the target protein. SUMOylation is a regulated process controlling protein activity, allowing rapid response to environmental signals. SUMOylation influences nerve function by regulating ion channels, transporters, and signalling molecules [18,22,23,24]. The SUMOylation of certain proteins protects peripheral nerves from degeneration and oxidative stress [25,26]. However, deSUMOylation in sensory neurons during type 1 diabetes can cause metabolic imbalance and nerve damage [18]. Our previous study on type 1 diabetes demonstrated that SUMOylation positively regulates key metabolic enzymes, including Glyceraldehyde-3-phosphate dehydrogenase (GAPDH), citrate synthase, and transketolase-like protein-1 (TKTL1), within the Krebs cycle. Loss of SUMOylation suppresses their activity, leading to glucose metabolite diversion towards the polyol pathway [18]. Consequently, this metabolic shift induces oxidative stress, the formation of advanced glycation end products (AGEs), and energy depletion [27]. However, the impact of SUMOylation on oxidative stress modulation in neuronal tissue remains poorly understood, and nothing is known as yet about the sequence of pathophysiological events involving SUMOylation in peripheral nerves in type 2 diabetes. Through the use of genetically modified mice lacking SUMOylation in peripheral sensory neurons, this study elucidates the neuroprotective role of SUMOylation via its key role in generating and regulating oxidative stress in type 2 diabetes.

## 2. Materials and Methods

### 2.1. Cell Culture

Human embryonic kidney cells (HEK 293, obtained from ATCC, identifier CVCL_0045) were cultured in Dulbecco’s modified Eagle medium (DMEM, Gibco NewYork, USA supplemented with 10% fetal bovine serum (FBS), 4 mM L-glutamine, 100 µg/mL streptomycin, and 100 U/mL penicillin. The cells were grown at 37 °C and 5% CO_2_ in a humidified incubator. The cells were split 24 h prior to transfection to achieve 70% confluency at the time of transfection. HEK293 cells were transfected with 20 µg of plasmid DNA (pSENP1 or control plasmid) using the calcium chloride transfection method described in [28]. Briefly, 20 µg of DNA was mixed with 48 µL of CaCl_2_, 510 µL of 1× BBS (pH 6.9), and 441 µL of H_2_O. The mixture was incubated at 37 °C for 10 min and subsequently added dropwise to the cells in 6 mL of DMEM without supplements. After 8 h, the medium was changed to full DMEM. Twenty-four hours post-transfection, the cells were harvested for Western blot analysis, or assays for activity of malate dehydrogenase (MDH) were performed. 

Dorsal root ganglion (DRG) cultures were made from SNS-Ubc9^−/−^ and Ubc9^fl/fl^ mice. DRG were isolated from 4–5-week-old mice and incubated in a mixture of digestive enzymes containing trypsin, collagenase, and DNase at 37 °C for 30 min. The neurons were enriched from non-neuronal cells using the Percol purification procedure described in Simonetti et al. [29]. Briefly, 9 mL of Percol was mixed with 1 mL of 10X PBS, and a gradient was formed in a 15 mL falcon tube. The lower phase consisted of 1.5 mL 35% Percol solution in pure DMEM, and it was topped carefully with 2.5 mL of a 25% Percol solution in DMEM. The resuspended cells were added carefully on top of the upper Percol phase, and everything was centrifuged for 15 min at 150 g. The uppermost 2 mL were carefully removed, and the remaining cells were washed twice with DMEM supplied with 10% FBS. DRG neurons were plated on poly-L-lysine-coated culture plates for 24 h before MDH activity assay or immunoblotting studies.

### 2.2. Immunoprecipitation and Western Blot Analysis

Immunoprecipitation was performed either from HEK cells or mouse DRG tissues, as described previously [30]. Briefly, the cells or tissue material were lysed in 1% sodium dodecyl sulphate (SDS) RIPA buffer supplemented with 200 mM N-ethylmaleimide (NEM), 0.5 mM DTT, and protease inhibitor cocktail. The lysate was sonicated for 15 s and then heated at 60 °C for 30 min. The lysate was centrifuged for 10 min at 13,000× *g* and the supernatant was collected. The collected lysate was further diluted to 0.1% SDS concentration and subjected to immunoprecipitation using either 5–7 µg of anti-SUMO1 (DSHB, cat no. 21C7) or IgG control antibody (Sigma Aldrich, St. Louis, MO, USA, cat no. I5381) and 25 µL of either Myc-Trap (Chromotek, Planegg, Germany, cat no. ytma) or control-Trap beads (Chromotek, cat no. bab) and incubated for 24 h at 4 °C or for 2 h for the Trap beads. In the case of OxPhos and PKG1, DRG tissue was lysed in 0.1% SDS RIPA buffer without DTT [31]. The lysate and immunoprecipitated proteins were resolved on SDS-PAGE, transferred to a nitrocellulose membrane, and blotted with the desired primary antibodies (Oxphos: Thermo Fischer, cat no. 45-8099, 1:1000; PKG1: Enzo Life Science, cat no. ADI-KAP-PK005, 1:750; MDH2: Cell Signaling, cat no. #8610, 1:500; Myc, Origene, cat no. TA150121, 1:1000), and, accordingly, the appropriate Alexa Fluor-conjugated secondary antibodies (either Alexa Fluor 488 anti-mouse IgG, Thermo Fischer, cat no. A32766, 1:2500 or Alexa Fluor 594 anti-rabbit IgG, Thermo Fischer, cat no. A32754, 1:2500) using standard Western blotting protocol [32]. The desired protein band was detected using the iBright 1500 imaging system (Thermo Fisher, Karlsruhe Germany). All experiments were repeated at least three times.

### 2.3. MDH Activity Assay

The assay for determining MDH activity was performed either with transfected HEK cells or from the DRG tissue isolated from SNS-Ubc9^−/−^ and Ubc9^fl/fl^ mice using a commercial kit as per manufacturer’s protocol (Sigma Aldrich, cat no. MAK196). Briefly, cells and DRG tissue were lysed in 1× reaction buffer from the kit using mild sonication. The lysate was centrifuged at 13,000× *g* for 10 min at 4 °C and the supernatant was transferred into a new tube. MDH activity was determined in supernatants by generating a product with absorbance at 450 nm proportional to the enzymatic activity present following the manufacturer’s instructions. Furthermore, the same lysate samples were used in Western blot analysis to determine the expression levels of MDH2 and actin. The MDH2 protein levels in each sample were normalised over control protein actin prior to the final quantification of MDH activity. The analyses were derived using 4–7 biological samples with 2–3 technical replicates in each experiment.

### 2.4. Animal Experiments

Age-matched 8–10-week-old SNS-Ubc9^−/−^ and Ubc9^fl/fl^ mice were used for all experiments. Two to four littermates were housed in individually ventilated enriched cage-rack systems in a temperature- and humidity-controlled room with an inverse light/dark cycle (12/12). All animals had free access to water and food. Mice of both sexes were used in all of the experiments in equal numbers. All animal experimental procedures were approved by the Regierungspräsidium Karlsruhe: Abteilung 3, Landwirtschaft, Ländlicher Raum, Veterinär und Lebensmittelwesen’, approval numbers 35-9185.81/G-133/19, All animal experiments followed institutional guidelines and were conducted in accordance with German Law, which regulates animal welfare and the protection of animals used for scientific purposes (TierSchG, TierSchVersV).

### 2.5. Sensory-Neuron-Specific Knockout Mice

To investigate the role of SUMOylation, we employed mice lacking Ubc9 protein (SNS-Ubc9^−/−^ mice), the only known E2-conjugating enzyme in the SUMOylation [33,34,35] pathway. Loss of Ubc9 in DRG sensory neurons abolishes the SUMOylation of proteins. The generation of SUMOylation-deficient SNS-Ubc9^−/−^ was previously described by Agarwal et al. [18]. Briefly, mice carrying homozygous alleles of the Ubc9 gene (Ubc9^fl/fl^ mice) were mated with mice expressing the Cre recombinase under the control of the Na_v_ 1.8 promoter (SNS-Cre mice) [36]. Homozygous SNS-Ubc9^−/−^ and control Ubc9^fl/fl^ mice were used for all experiments.

### 2.6. Rodent Model of Type 2 Diabetes

Diabetes was induced in mice via a high-fat diet (HFD) comprising 45% kcal as saturated fat from Lard (Ssniff, cat no. D12451) [37]. Five weeks post-HFD, three low doses of freshly prepared streptozotocin (STZ) (40 mg/kg) dissolved in citrate buffer (50 mM, pH 4.5) were intraperitoneally (i.p) injected to establish the non-genetic model of type 2 diabetes, as previously characterized [38]. Behavioural experiments were conducted at 10, 17, 23, and 30 weeks post-HFD. Glucose tolerance tests (GTT) were performed, involving a 24 h fast followed by i.p injection of 4.44 µL of 45% glucose solution per gram of body weight. Blood glucose levels were measured using a Roche Accu-Chek glucometer before and 15, 30, 60, and 120 min after glucose administration. Blood glucose values were measured in mg/dL and the area under the curve was calculated.

### 2.7. Behavioural Measurements

All behavioural experiments were performed with awake, unrestrained, and age-matched mice. Both male and female mice were included in behavioural recordings. The experimenter was blinded for both the gender and treatment in the experimental groups. Mice were acclimatised to the recording setup twice a day for four days before the recording. Mechanical sensitivity was measured using Von Frey filaments at 10, 17, 23, and 30 weeks post-HFD. Mice were placed on the elevated metal grid, and Von Frey monofilaments exerting a specific force of 0.07, 0.16, 0.4, 0.6, 1.0, 1.4, 2.0, and 4.0 g were tested on the plantar hind paw. Each filament was applied five times at a time interval of 3–4 min on the hind paw. Sixty percent response frequency was calculated at 10, 17, 23, and 30 weeks post-HFD, as described previously [29,39,40]. Thermal sensitivity was measured by recording paw withdrawal latency upon the application of an infrared heat source on the plantar surface of the hind paw with an IR of 40, as described previously [40]. Mice were acclimatised for 30 min prior to recording. A twenty-second cut-off was set to prevent any heat-induced injury. Each paw was tested thrice, and 5 min time intervals separated two consecutive heat applications. 

### 2.8. Immunohistochemistry

Immunohistochemistry was performed on skin biopsies of the plantar surface of the hind paws from the diabetic and non-diabetic SNS-Ubc9^−/−^ and Ubc9^fl/fl^ mice as basal or 30 weeks post-HFD, as previously described [39,41]. Briefly, the mice were perfused with 4% Paraformaldehyde (PFA), and plantar skin was dissected and post-fixed with 4% PFA for 24 h at 4 °C. The biopsy samples were then incubated in 30% sucrose in 1× phosphate buffer saline (PBS), and 16 µm cryosections were made. The sections were permeabilised with 0.5% PBST, followed by washing, and blocked with 7% horse serum in 1× PBST. Next, the sections were incubated with either anti-calcitonin gene-related peptide (anti-CGRP Sigma-Aldrich, cat. no. C8198, 1:1000) or anti-protein gene product 9.5 (anti-PGP9.5, Dako, Glostrup, Denmark, cat no. Z5116 1:1000) primary antibodies for 48 h at 4 °C. After that, the sections were washed and incubated with Alexa Fluor-488-conjugated secondary antibodies (Thermo Fischer, cat no. A32766, 1:700) for 1 h at room temperature. Finally, the sections were washed and mounted in Mowiol 4–88 (Carl Roth, Karlsruhe, Germany, cat no. 0713.1). Fluorescence images were obtained using a Leica SP8 confocal microscope using identical image-acquiring conditions across different sections and test groups. Acquired images were analysed using Image J software (version 1.8.0_172), as described previously [41]. The epidermal area in the hind paw skin biopsies was marked, and the total fluorescence density over the marked epidermal area was calculated after subtracting the background signal from each test sample [18,42].

### 2.9. Metabolomics

#### 2.9.1. Sample Extraction

Sciatic nerves from diabetic and non-diabetic SNS-Ubc9^−/−^ and Ubc9^fl/fl^ mice were isolated for metabolomics analysis, as previously described in [18]. Mice were deeply anesthetised, and sciatic nerve tissue was isolated and frozen in liquid N_2_. Frozen sample material was extracted with 190 µL 100% methanol containing 10 µL 0.02 mg/mL Ribitol for 15 min at 70 °C with vigorous shaking. After the addition of 100 µL 100% chloroform, samples were shaken at 37 °C for 5 min. To separate polar from organic phases, 200 µL of HPLC-grade water was added, and samples were centrifuged for 10 min at 11,000× *g* while avoiding the interphase containing cellular debris. In total, 350 µL of the polar (upper) phase was transferred to a fresh glass vial and dried using a vacuum concentrator (Eppendorf Concentrator Plus, Eppendorf, Hamburg, Germany) without heating. 

#### 2.9.2. Derivatisation (Methoximation and Silylation)

Sequential online methoximation and silylation reactions were performed using an MPS autosampler (Gerstel, Mülheim Ruhr, Germany). Methoximation was achieved by adding 20 µL of 20 mg/mL methoxyamine hydrochloride (Sigma 226904) in pyridine (Sigma 270970) and incubating at 37 °C for 90 min in a Gerstel MPS Agitator Unit (250 rpm). For sialylation reactions, 45 µL of N-Methyl-N-(trimethylsilyl)trifluoroacetamide (MSTFA; Sigma 69479) was added, and samples were incubated at 37 °C for 30 min with gentle shaking. Before injection, samples were incubated at room temperature for 45 min. 

#### 2.9.3. Gas Chromatography/Mass Spectrometry (GC/MS) Analysis

For GC/MS analysis, a GC-ToF system was used consisting of an Agilent 7890 Gas Chromatograph (Agilent, Santa Clara, CA, USA) fitted with an Rxi-5Sil MS column (30 m × 0.25 mm × 0.25 µm; Restek Corporation, Bellefonte, PA, USA) coupled to a Pegasus BT Mass Spectrometer (LECO Corporation, St. Joseph, MI, USA). The GC was operated with an injection temperature of 250 °C, and a 1 µL sample was injected in split less mode. The GC temperature program started with a 1 min hold at 40 °C followed by a 6 °C/min ramp up to 210 °C, a 20 °C/min ramp up to 330 °C, and a bake-out at 330 °C for 5 min using Helium as carrier gas with constant linear velocity. The ToF mass spectrometer was operated with ion source and interface temperatures of 250 °C, a solvent cut time of 9 min, and a scan range (*m*/*z*) of 50–600 with an acquisition rate of 17 spectra/second. ChromaTof v5.50 software (LECO Corporation, St. Joseph, MI, USA) was used for data processing. 

### 2.10. Statistics

Our sample sizes are similar to previous studies [18,43,44], where we determined the sample size using G-power analysis for specific behavioural and morphological or molecular experiments. All data shown are represented as mean ± standard error of the mean (SEM). When comparing multiple groups and variables, a two-way Analysis of Variance (ANOVA) test followed by a post hoc Bonferroni’s test were performed. When comparing two groups of data, a Mann–Whitney test was used. The exact statistical test employed for each data set is indicated in the figure legends. All statistical tests were performed in GraphPad, GraphPad Software Inc., San Diego, CA, USA. A *p*-value of ≤ 0.05 was considered significant.

## 3. Results

### 3.1. Mice Lacking SUMOylation, Specifically in DRG Sensory Neurons, Exhibit Accelerated DPN

SNS-Ubc9^−/−^ and corresponding control mice were used to investigate the role of SUMOylation in DPN development longitudinally, as shown in the experimental scheme in Figure 1a. SNS-Ubc9^−/−^ and Ubc9^fl/fl^ mice showed similar glucose intolerance caused by insulin resistance and insufficient insulin secretion at 16 and 29 weeks post-HFD (Figure 1b), showing that the development of type 2 diabetes was comparable in both genotypes. In behavioural assays, SNS-Ubc9^−/−^ and Ubc9^fl/fl^ mice were indistinguishable at the early time point post-HFD as compared to the basal state; however, at 30 weeks, mutant mice developed mechanical (Figure 1c) and thermal (Figure 1d) hypoalgesia, while control mice did not. 

To correlate the behavioural changes with alteration in nerve morphology, we performed immunostaining using pan-nerve antigen PGP9.5 (Protein gene product 9.5) and nociceptive peptidergic marker (Calcitonin gene-related peptide, CGRP) on the hind paw skin sections from SNS-Ubc9^−/−^ and Ubc9^fl/fl^ mice. Naïve SNS-Ubc9^−/−^ and Ubc9^fl/fl^ mice were indistinguishable (Figure 2a–d), and both genotypes displayed reduced epidermal nerve fibre density at 30 weeks post-HFD compared to the basal state. However, SUMO-deficient mice exhibited more severe nerve fibre loss than controls (Figure 2a,c; quantitative summary in Figure 2b,d).

### 3.2. SUMOylation Modulates Reactive Oxygen Species (ROS) Levels in DRG Neurons and the Nerve Ending in Paw Skin of Mice with Type 2 Diabetes

Hyperglycaemia elevates ROS levels in neurons due to increased ATP demand and electron escape from the respiratory chain [45]. The role of SUMOylation in modulating ROS levels remains unclear. We measured the oxidative state in DRG neurons of SNS-Ubc9^−/−^ and Ubc9^fl/fl^ mice under basal and diabetic conditions by assessing cGMP-dependent PKG1 dimer/monomer ratios, which serve as an ROS sensor in neurons [31]. Both genotypes showed similar PKG1 levels in basal states. However, the diabetic state increased the dimer/monomer ratio for both (Figure 3a). Notably, SUMO-deficient mice exhibited a significantly higher increase in the ratio compared to controls in the diabetic state, indicating SUMOylation’s role in impairing ROS generation during glucose homeostasis (quantification in Figure 3b). We investigated if inflated ROS levels extend to skin nerve endings. Mito Tracker Red CM-H2XROS dye was injected intraplantar in SNS-Ubc9^−/−^ and Ubc9^fl/fl^ mice at basal or 30 weeks post-HFD, whereby detection of fluorescence indicates ROS generation and mitochondrial dysfunction [42]. We performed immunostaining of PGP9.5 and quantified ROS levels within dermal PGP9.5-positive endings. Basal states showed no distinction between SNS-Ubc9^−/−^ and Ubc9^fl/fl^ mice. Both genotypes had increased ROS with diabetes progression at 30 weeks post-HFD (Figure 3c, quantification in Figure 3d). Notably, SUMO-deficient SNS-Ubc9^−/−^ mice had significantly higher ROS levels than Ubc9^fl/fl^ controls at 30 weeks post-HFD (Figure 3c, quantification in Figure 3d). These findings suggest deSUMOylation inflates ROS levels in DRG neuron soma as well as in peripheral nerve fibres in the skin. Next, we studied the impact of SUMOylation on cellular defence against oxidative stress using mass spectrometry to measure glutathione (GSH) and cysteine levels in DRG of SNS-Ubc9^−/−^ and control Ubc9^fl/fl^ mice at basal and 30 weeks post-HFD. GSH levels were significantly reduced in SUMO-deficient mice compared to controls at basal state (Figure 3e). During diabetes progression, both genotypes showed increased GSH levels compared to the basal state, but the increase was significantly lower in SNS-Ubc9^−/−^ than in controls (Figure 3e). Cysteine levels were lower in SNS-Ubc9^−/−^ than control Ubc9^fl/fl^ mice at both basal and 30 weeks post-HFD (Figure 3f), suggesting compromised detoxification potential in SUMO-deficient mice.

### 3.3. Impact of SUMOylation on the Respiratory Chain

We then investigated how SUMOylation affects the functionality of the mitochondrial respiratory chain, thus potentially leading to increased ROS levels when SUMOylation is absent. We measured respiratory chain complex levels (I-V) in DRG of SNS-Ubc9^−/−^ and control Ubc9^fl/fl^ mice at basal state and 30 weeks post-HFD. At basal state, both genotypes showed similar levels of complexes I-V. However, during diabetes progression, SUMO-deficient SNS-Ubc9^−/−^ mice exhibited reduced levels of complexes II, III, IV, and V compared to control Ubc9^fl/fl^ mice (Figure 3g, quantification in Figure 3h). These findings highlight the vital role of SUMOylation in maintaining respiratory chain integrity, as its deficiency may lead to respiratory chain dysfunction and excessive ROS levels.

### 3.4. Malate Dehydrogenase as a SUMOylation Target

We explored factors influencing ROS levels, including the NADH/NAD+ redox couple balance. Malate dehydrogenase 1 (MDH1) and mitochondrial malate dehydrogenase 2 (MDH2) regulate this ratio [46]. In a previous study, we identified MDH2 as a SUMOylation target in mouse dorsal root ganglia sensory neurons using LC-MS/MS [18]. We validated MDH2 as a SUMOylation target in vitro by overexpressing Myc-tagged-MDH2 in HEK cells and performing immunoprecipitation studies. Immunoprecipitation with anti-SUMO1 antibody and immunoblotting with anti-Myc antibody confirmed MDH2 mono-SUMOylation (Figure 4a, arrow). A band shift of 15–20 kDa was observed (Figure 4a, arrowhead for unSUMOylated MDH2 and arrow for SUMOylated MDH2), indicative of mono-SUMOylation. Immunoprecipitation with anti-SUMO1 antibody from DRG tissue and immunoblotting with anti-MDH2 antibody further verified MDH2 SUMOylation (Figure 4b). Specificity controls using IgG for immunoprecipitation did not show any bands, supporting MDH2 SUMOylation specificity. We assessed the impact of de-SUMOylation on MDH2 activity in HEK cells and DRG tissue. Transfecting HEK cells with SENP-1, a SUMO-specific isopeptidase, increased MDH2 activity compared to control transfection (Figure 4c) without affecting MDH2 protein levels (Figure 4e). Similarly, in SNS-Ubc9^−/−^ mice, deletion of SUMOylation led to increased MDH2 activity compared to control mice (Figure 4d). However, the levels of MDH2 proteins were unchanged in mutant versus control mice (Figure 4f). This increased MDH2 activity reduced ROS production, potentially serving as a protective mechanism against inflated ROS levels resulting from a dysfunctional respiratory chain in SUMO-deficient mice.

Using LC-MS and MS, we assessed lipid levels in sciatic nerve tissue from SNS-Ubc9^−/−^ and Ubc9^fl/fl^ mice in basal and diabetic states. Linoleic acid and eicosatrienoic acid are crucial omega-6 and omega-3 fatty acids that play vital roles in maintaining neuronal membrane integrity and fluidity, which are essential for proper neuron function [47]. Linoleic acid levels increased with disease progression in wildtype diabetic mice, while SUMO-deficient mice consistently exhibited significantly lower linoleic acid levels post30 weeks of HFD compared to control mice (Figure 5a). Similarly, eicosatrienoic acid levels were significantly lower in diabetic SNS-Ubc9^−/−^ mice compared to controls (Figure 5b). Reduced levels of these acids may render the neuronal membrane vulnerable to damage in diabetes. We also assessed stearic acid, oleic acid, and tricosanoic levels. Stearic acid and tricosanoic levels increased with diabetes progression but showed no difference between SNS-Ubc9^−/−^ and Ubc9^fl/fl^ mice (Figure 5c,d). Notably, oleic acid, known for its neuroprotective properties and antioxidant effects [48], increased in Ubc9^fl/fl^ mice after 30 weeks of HFD; this increase was compromised in SNS-Ubc9^−/−^ mice (Figure 5e). Thus, SUMOylation-deficient neurons lack the neuroprotective benefits of oleic acid, which may underlie the increased oxidative stress and nerve damage we observed in SUMOylation-deficient mice under diabetic conditions. Our findings thus point to a high level of significance for SUMOylation in exerting neuroprotective functions in diabetes, offering potential insights for therapeutic interventions.

## 4. Discussion

Diverse post-translational modifications have been reported in late diabetic complications. However, the molecular pathogenesis of diabetic peripheral neuropathy (DPN) remains poorly understood. Notably, SUMOylation has been linked to neurodegenerative diseases, like Alzheimer’s and Huntington’s disease [16,49,50]. Conversely, its neuroprotective role has been observed in type 1 diabetes [18,19], stroke, and brain ischemia [17]. Our previous research using mice lacking SUMOylation in sensory neurons demonstrated its protective function against metabolic damage and sensory loss induced by hyperglycaemia in type 1 diabetes [18]. However, the effects of SUMOylation on energy generation and ROS modulation in type 2 diabetes have not been investigated. This study fills this gap by revealing that SUMOylation influences the respiratory chain, thereby modulating ROS levels in peripheral sensory neurons under metabolic stress in a type 2 diabetes model. DeSUMOylation in sensory neurons accelerated sensory loss in the type 2 diabetes model induced by a high-fat diet, which is consistent with SUMOylation’s neuroprotective role observed in type 1 diabetes. These findings support the previously identified neuroprotective role of SUMOylation in type 1 diabetes. Although the pathophysiology of type 1 and 2 diabetes differ, our findings show that SUMOylation protects the nerve tissue from metabolic-dysregulation-induced damage. 

A number of interesting mechanistic processes may contribute to this outcome. SUMOylation influences glucose metabolism, including insulin signalling, gluconeogenesis, glycolysis, and glucose uptake. SUMOylation of insulin receptor substrate (IRS-1), Glyceraldehyde-3-phosphate dehydrogenase (GAPDH), pyruvate kinase M2 (PKM2), and glucose transporter 4 (GLUT4) promote glucose homeostasis [18,51]. DeSUMOylation of metabolic enzymes suppresses Kreb’s cycle and shuttles the glucose flux to harmful aldose reductase polyol pathways [18]. Here, we explored if this SUMOylation-dependent metabolic imbalance is associated with the modulation of ROS levels. Our data of ROS measured in peripheral sensory nerve tissue indicate that the excessive ROS levels are not limited to the soma of peripheral sensory neurons. It is possible that inflated levels of ROS lead to oxidative stress in the soma, which spreads to the axonal ending, which may cause cellular damage and accelerated neuropathy. The reduced levels of glutathione and cysteine in the sciatic nerve of SUMOylation-deficient diabetic mice indicate compromised antioxidant capability at the distal ends of the nerves. Numerous research studies have reported a decline in the activity of GSH in liver and kidney tissues in the context of diabetes [52,53]. Conversely, an elevation in GSH activity has been noted in peripheral neuronal tissue in various models of neuropathic pain [54,55]. In our own observations, we found heightened GSH levels in Ubc9^fl/fl^ diabetic mice when compared to non-diabetic control mice, but this increase was reduced in SUMOylation-deficient mice. Consequently, our findings suggest that SUMOylation serves as a protective mechanism against the loss of redox balance, the absence of which contributes to neuronal dysfunction. 

Hyperglycaemia, mitochondrial membrane potential disruption, or dysfunction of ETS complexes contribute to excessive ROS production [56]. We explored the potential role of SUMOylation in influencing the stability of ETS complexes and whether alterations in the SUMOylation status within DRG may impact ETS functionality, subsequently affecting the generation of ROS. We observed the reduced levels of complex III, IV, and V in SUMOylation-deficient diabetic mice. ETS complex III is associated with electron transfer from ubiquinol to cytochrome c. Impaired complex III functions or levels resulted in electron leakage and increased ROS production [57,58]. Complex IV is the final electron acceptor in the ETS; it transfers electrons from cytochrome c to oxygen, thus forming water. Reduced levels or compromised complex IV contribute to electron leakage, incomplete reduction of oxygen, and, thus, ROS production [59]. Although Complex V does not directly participate in ROS production, a decrease in its activity or levels can lead to a decrease in ATP synthesis. This energy deficit can compromise the cellular antioxidant defence systems, reducing the cell’s ability to counteract ROS and increasing oxidative stress. It could be postulated that SUMOylation may modulate the expression, trafficking, or assembly of ETS complex under diabetic conditions in sensory neurons. Under diabetic conditions, there is a need to have enhanced and efficient ETS function that is linked to increased expression and import of ETS complex. The transcription factors Nuclear respiratory factor-1 and -2 (NRF1 and NRF2) and cofactors, like Peroxisome proliferator-activated receptor gamma coactivator 1-alpha (PGC-1α), regulate ETS gene expression and mitochondrial biogenesis. The activity and these proteins are shown to be directly or indirectly modulated by SUMOylation [60,61]. DeSUMOylation may cause reduced expression of ETS complex proteins under diabetic conditions. SUMOylation likely impacts mitochondrial dynamics and function by modulating protein import. Heat shock proteins Hsp70 and Hsp90 are vital for mitochondrial import complex stability and are known to be SUMOylated [26,62]. DeSUMOylation in diabetes might compromise Hsp70 and Hsp90 function, leading to reduced mitochondrial import and impaired ETS complex assembly.

Indeed, the reduced levels of the ETS complex we observed in diabetic SNS-Ubc9^−/−^ as compared to control Ubc9^fl/fl^ mice demonstrate that SUMOylation is a crucial biological process required to maintain the physiological functions of the respiratory chain under pathophysiological conditions. An inefficient ETS chain could also result from an imbalanced NADH/NAD+ ratio [63]. NADH is a critical cofactor in the ETS for ATP production, and efficient NADH production by MDH2 is important for maintaining a balanced redox state and ROS levels [64]. In our prior mass spectrometry screening, we identified MDH2 as a target of SUMOylation [18]. However, we did not conduct an in-depth investigation into the implications of SUMOylation on MDH2 activity and its contribution to the maintenance of respiratory chain functionality. In this report, we present evidence indicating that MDH2 indeed undergoes SUMO modification, and this modification plays a pivotal role in modulating its functional properties. Our data indicate that deSUMOylation of MDH2 enhances its activity in vitro and in vivo and may result in a higher NADH/NAD+ ratio. It implies that NADH production is not impaired in the absence of SUMOylation. In contrast to other metabolic enzymes, such as GAPDH, whose activity is SUMOylation-dependant [18], the increased activity of MDH2 following SUMOylation is likely to maintain ETS functionality and energy supply and thereby help to overcome cellular stress and damage caused by hyperglycaemia. 

Our results show that diabetic neuropathy is associated with mitochondrial dysfunction, and it is regulated by SUMOylation. SUMOylation-deficient mice exhibit substantial impairment in the mitochondrial oxidative phosphorylation complexes III, IV, and V within the respiratory chain under diabetic conditions. However, establishing causal links between mitochondrial dysfunction, oxidative stress, and sensory loss in diabetes remains a crucial area of investigation. Exploring the potential therapeutic role of mitochondria-targeted antioxidants, like MitoTEMPOL [65] or ROS scavengers, could unveil promising avenues for treating DPN. Our findings also highlight the impact of deSUMOylation on MDH2, leading to increased enzymatic activity. Investigating MDH2 inhibitors, such as compound 7 [66] and LW6 [67], may elucidate MDH2’s therapeutic potential. 

From a therapeutic viewpoint, enhancing SUMOylation pathway enzymes in peripheral neurons could potentially prevent or reverse diabetic neuropathy. Presently, there are no highly specific drugs activating these enzymes. However, certain compounds, like Gingko extract, N106 (specific for SUMO-activating enzyme E1 ligase) [68,69], or SENP2 inhibitors, like 1,2,5-Oxadiazole [70], can modulate SUMOylation and may be beneficial in diabetes. An appealing strategy involves employing a targeted gene delivery system [71]) to restore deficient SUMOylation enzymes within peripheral neurons localized in the dorsal root ganglia. Exploring this avenue holds significant promise for evaluating the therapeutic potential of modulating SUMOylation in the context of diabetic neuropathy, showcasing a compelling direction for future research endeavours. Although our study focuses on an HFD-induced type 2 diabetes model, future research with genetically modified mice, like leptin-deficient db/db mice [72], may reinforce the significance of the SUMOylation pathway in DPN onset and progression.

## 5. Conclusions

Based on our results, we conclude that SUMOylation is an essential neuroprotective mechanism in sensory neurons in type 2 diabetes, the loss of which results in oxidative stress at nerve terminals and suppresses the efficiency of the respiratory chain. Consequently, this impairment contributes significantly to nerve damage in DPN. 

## Figures and Tables

**Figure 1 cells-12-02511-f001:**
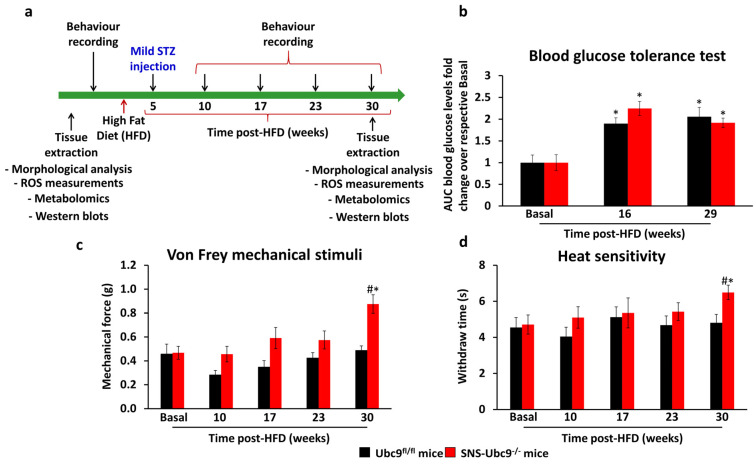
**Analysis of hyperglycaemia and behavioural measurements of mice lacking SUMOylation in peripheral nociceptive neurons or control mice in a type 2 model of diabetes in a longitudinal study spanning 30 weeks.** (**a**) Experimental scheme for the used model to induce type 2 diabetes and the time points testing for mechanical sensitivity (von Frey) and heat sensitivity (Hargreaves). (**b**) Analysis of blood glucose tolerance tests at different stages throughout type 2 diabetes. The area under the curve was analysed. N = 6 mice/genotype; two-way ANOVA test was performed. * *p* ≤ 0.05 compared to basal. Data are expressed as mean ± SEM. (**c**,**d**) Course of DPN-associated changes in the response threshold and thermal latency before and after inducing type 2 diabetes. N = 6 mice/genotype; repeated two-way ANOVA test was performed. * *p* ≤ 0.05 compared to basal, # *p* ≤ 0.05 compared to the control group. Data are expressed as mean ± SEM.

**Figure 2 cells-12-02511-f002:**
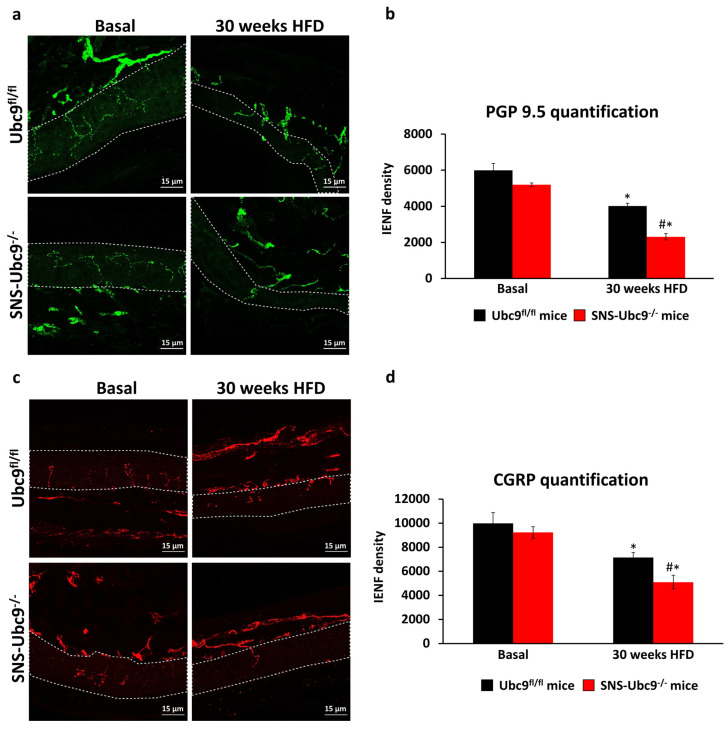
**Quantitative and morphological analysis of DPN-associated loss of peripheral nerve endings in the plantar epidermis in SNS-Ubc9^−/−^ and Ubc9^fl/fl^ mice.** (**a**,**c**) Identification of intraepidermal nerve fibre endings via immunofluorescence staining for PGP9.5 (**a**) or CGRP in the paw skin from SNS-Ubc9^−/−^ and Ubc9^fl/fl^ mice at basal stage and after 30 weeks of type 2 diabetes. The dotted white line represents the epidermal layer. Scale bar corresponds to 15 µm. (**b**,**d**) Quantification of the reduction of intraepidermal nerve fibre density after 30 weeks of diabetes in SNS-Ubc9^−/−^ and Ubc9^fl/fl^ mice. N = 3–4 mice/genotype; two-way ANOVA test was performed. * *p* ≤ 0.05 compared to basal, # *p* ≤ 0.05 compared to the control group. Data are expressed as mean ± SEM.

**Figure 3 cells-12-02511-f003:**
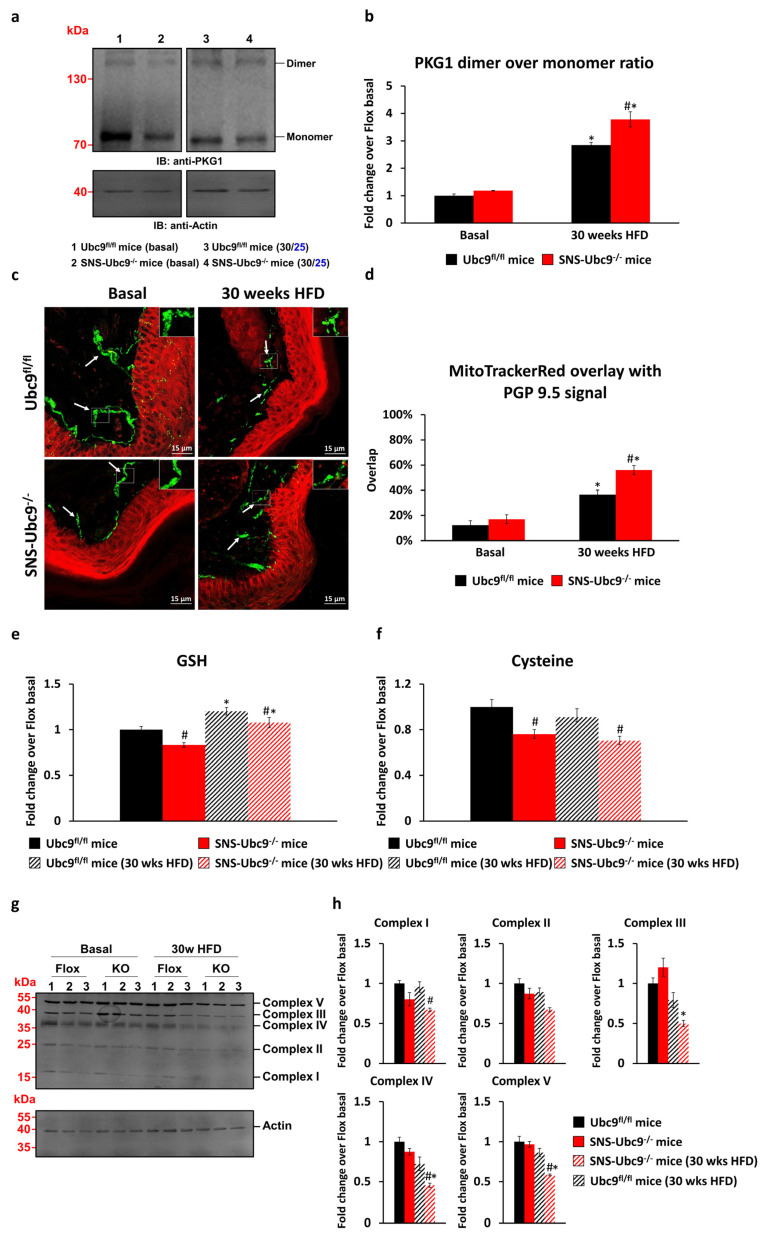
**Quantitative analysis of ROS measurements and changes in the electron transport chain in vivo in the DRGs and the paw skin.** (**a**) Western blot analysis of PKG1α expression in SNS-Ubc9^−/−^ and Ubc9^fl/fl^ mice at basal stage and after 30 weeks of type 2 diabetes. (**b**) Analysis of band intensities using densitometry; dimerized PKG1α was quantified over monomeric PKG1α. N = 3 mice/genotype; two-way ANOVA test was performed. * *p* ≤ 0.05 compared to basal, # *p* ≤ 0.05 compared to the control group. Data are expressed as mean ± SEM. (**c**) Representative pictures of MitoTracker Red staining overlapping with PGP9.5 immunofluorescence stained nerve fibres in the dermis in SNS-Ubc9^−/−^ and Ubc9^fl/fl^ mice at basal stage and after 30 weeks of type 2 diabetes after injection into the paw. Scale bar corresponds to 15 µm. Arrows indicate examples for analysed fibres in the dermis.(**d**) Quantitative analysis of the MitoTracker Red overlap in PGP9.5 stained nerve fibres. N = 3–4 mice/genotype; two-way ANOVA test was performed. * *p* ≤ 0.05 compared to basal, # *p* ≤ 0.05 compared to the control group. Data are expressed as mean ± SEM. (**e**) Quantitative analysis of GSH levels in the sciatic nerve from SNS-Ubc9^−/−^ and Ubc9^fl/fl^ mice at basal stage and after 30 weeks of type 2 diabetes. N = 5–6 mice/genotype; two-way ANOVA test was performed. * *p* ≤ 0.05 compared to basal, # *p* ≤ 0.05 compared to the control group. Data are expressed as mean ± SEM. (**f**) Quantitative analysis of cysteine levels in the sciatic nerve from SNS-Ubc9^−/−^ and Ubc9^fl/fl^ mice at basal stage and after 30 weeks of type 2 diabetes. N = 5–6 mice/genotype; two-way ANOVA test was performed., # *p* ≤ 0.05 compared to the control group. Data are expressed as mean ± SEM. (**g**) Western blot analysis of the expression of different key proteins from the oxidative phosphorylation cascade in SNS-Ubc9^−/−^ and Ubc9^fl/fl^ mice at basal stage and after 30 weeks of type 2 diabetes. Complex I–NDUFB8, Complex II–SDHB, Complex III–UQCRC2, Complex VI–MTCO1, Complex V–ATP5AF1A. (**h**) Analysis of band intensities using densitometry. N = 3 mice/genotype; two-way ANOVA test was performed. * *p* ≤ 0.05 compared to basal, # *p* ≤ 0.05 compared to the control group. Data are expressed as mean ± SEM.

**Figure 4 cells-12-02511-f004:**
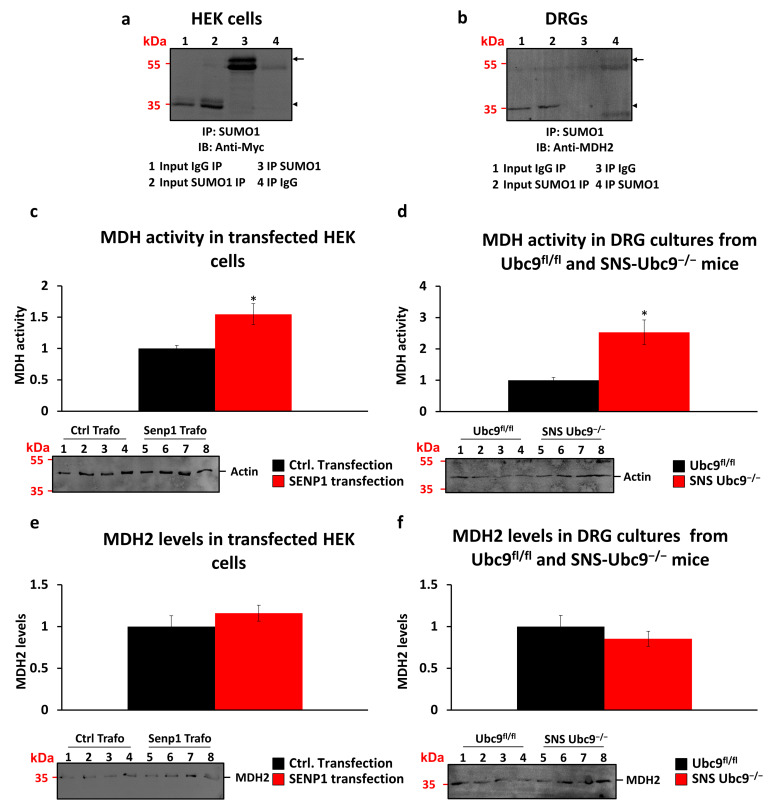
**MDH2 is a SUMOylation target and MDH activity is altered by the SUMOylation status.** (**a**) IP of artificial expressed Myc-tagged MDH2 in HEK cells and (**b**) native MDH2 from DRGs using an anti-SUMO1 antibody. The arrowhead indicates the unSUMOylated MDH2, while the arrow shows the SUMOylated MDH2. (**c**) MDH activity is increased in HEK cells transfected with a SENP1 expressing plasmid in comparison to control transfected HEK cells, while (**e**) MDH2 protein levels are not altered. N = 7 for SENP1 and control transfection; Mann–Whitney test was performed. * *p* ≤ 0.05. Data are expressed as mean ± SEM. (**d**) MDH activity is increased in DRG cultures obtained from 4-week-old SNS-Ubc9^−/−^ mice in comparison to the ones from Ubc9^fl/fl^ mice. (**f**) MDH2 protein levels are not altered between genotype. N = 4 mice/genotype; Mann–Whitney test was performed. * *p* ≤ 0.05. Data are expressed as mean ± SEM.

**Figure 5 cells-12-02511-f005:**
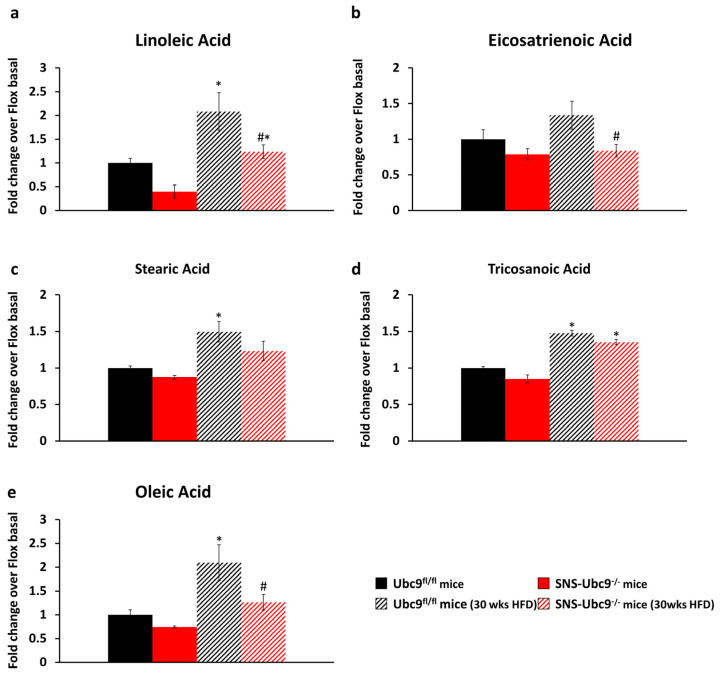
**Quantitative analysis of different fatty acid metabolites in the sciatic nerve from SNS-Ubc9^−/−^ and Ubc9^fl/fl^ mice at basal stage and after 30 weeks of type 2 diabetes.** Quantitative analysis of (**a**) Linoleic acid (**b**) Eicosatrienoic acid (**c**) Stearic acid (**d**) Tricosanoic acid (**e**) Oleic acid levels in the sciatic nerve from SNS-Ubc9^−/−^ and Ubc9^fl/fl^ mice at basal stage and after 30 weeks of type 2 diabetes. N = 5–6 mice/genotype; two-way ANOVA test was performed. * *p* ≤ 0.05 compared to basal, # *p* ≤ 0.05 compared to the control group. Data are expressed as mean ± SEM.

## Data Availability

Data are available upon request.

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
