# Peer review of "SUMOylation Modulates Reactive Oxygen Species (ROS) Levels and Acts as a Protective Mechanism in the Type 2 Model of Diabetic Peripheral Neuropathy"

_cells, 2023, doi:10.3390/cells12212511_

Round 1

Reviewer 1 Report

Dear author,

Thanks for submitting your research manuscript entitled "SUMOylation modulates reactive oxygen species (ROS) levels and acts as a protective mechanism in the type 2 model of diabetic peripheral neuropathy".

From my experience in the field, this manuscript is dealing with very important aspect of SUMOylation modulates reactive oxygen species (ROS) levels and acts as a protective mechanism in the type 2 model of diabetic peripheral neuropathy.

I recommend the revision of this manuscript (with minor comments) as it worth publishing and will add good information in the field of diabetic peripheral neuropathy.

Before giving my final comments and revising this manuscript, the author must address the following comments scientifically.
Reviewer  concerns:

Please find out the following comments:

·         The rationale and purpose behind selecting SUMOylation modulates reactive oxygen species (ROS) levels and acts as a protective mechanism in the type 2 model of diabetic peripheral neuropathy is unclear and need to reframe in introduction and discussion.

·         Updates old & outdated references.

·         Remove these types of vague words throughout the manuscript “ Diabetic peripheral neuropathy (DPN) a most common form of peripheral neuropathy and frequently affects the nerves in the extremities, such as the feet, legs and hands. Being a multifactorial disorder, the molecular mechanisms underlying diabetic neuropathy are complex and not fully understood” Mentioned only those which you observe, don’t crisp the science,

·         Major drawback is the lack of supporting references and incomplete experimental and paradigms.

·         A separate detail paragraph required here, to explain the experimental design with detail explanation in flow chart also.

·         How was the sample size determined? Ideally, a priori sample size calculation should be performed to determine the appropriate sample size.
- normality and variance homogeneity should be assessed across all groups of the same outcome variable and not individual experimental groups. If the data were not normally distributed or variance homogeneity was not met, nonparametric tests need to be performed.
Parametric data should be reported as mean +/- sd, while nonparametric data should be given/displayed as median and interquartile range. Longitudinal data should be analyzed using repeated measures tests.

·         Color expression pattern of all western blot images are not properly arranged. As well as require scale bar and annotation sign in each panel of figure 2a, 2c, 3 c. Without this, manuscript can’t proceed further.

·         All results are very poorly explained. Revised explanation for all as above mentioned instructions.

-          Results need more clarification and significant justification. Differentiating between the outcome and the discussion sections is quite difficult.

-     To address the outcome of results separately, avoiding the disease condition and maintaining physiological condition. How they correlate with the existing literature, it would be better if the author restructured to take a more critical approach of SUMOylation modulates reactive oxygen species (ROS) levels and acts as a protective mechanism in the type 2 model of diabetic peripheral neuropathy.

-     In the discussion and the conclusion, the aims, rationale, and future perspectives are not evident clearly in relation to in-vitro experimentation.
-     The discussion is usually unorganized at the beginning to address and evaluate all the observations at the end. It makes the results easier to contextualize and more straightforward to comprehend.

- Furthermore, a minimal critical analysis should be provided, along with current study limitations and the future perspective as separate paragraphs.

-          Need to revise the conclusion scientifically. Not accepted in its current form.

-          A detailed revision shortening, ordering and following the commented ideas could improve this paper.

-          Several typewriting mistakes are present and need correction. This reviewer remains at the entire disposal for the next version.

Moderate editing of English language required

Author Response

Responses to reviewers by Mandel et al.:

The authors sincerely thank both reviewers for their in-depth review and their helpful and constructive feedback. We were pleased to note the overall positive nature of the appraisals and have taken significant measures to address the critical issues. Point-by-point responses to all comments are given below and the main changes in the main text are marked in red colour.

Reviewer 1 comments

Dear author,

Thanks for submitting your research manuscript entitled "SUMOylation modulates reactive oxygen species (ROS) levels and acts as a protective mechanism in the type 2 model of diabetic peripheral neuropathy".

From my experience in the field, this manuscript is dealing with very important aspect of SUMOylation modulates reactive oxygen species (ROS) levels and acts as a protective mechanism in the type 2 model of diabetic peripheral neuropathy.

I recommend the revision of this manuscript (with minor comments) as it worth publishing and will add good information in the field of diabetic peripheral neuropathy.

Response: We thank the reviewer for this scholarly and in-depth review and are very pleased to note the reviewer’s recommendation of revision with minor comments.

Before giving my final comments and revising this manuscript, the author must address the following comments scientifically.

Reviewer concerns:

Please find out the following comments:

  • The rationale and purpose behind selecting SUMOylation modulates reactive oxygen species (ROS) levels and acts as a protective mechanism in the type 2 model of diabetic peripheral neuropathy is unclear and need to reframe in introduction and discussion.

Response: We thank the reviewer for pointing this out and we agree with the reviewer. We have now made the rationale clearer in the introduction section.

  • Updates old & outdated references.

Response: We have updated the references as per reviewer’s suggestion, particularly with respect to including newest review articles. Old references are given when it involves an original study, which is ethically the correct thing to do.

  • Remove these types of vague words throughout the manuscript “Diabetic peripheral neuropathy (DPN) a most common form of peripheral neuropathy and frequently affects the nerves in the extremities, such as the feet, legs and hands. Being a multifactorial disorder, the molecular mechanisms underlying diabetic neuropathy are complex and not fully understood” Mentioned only those which you observe, don’t crisp the science,

Response: We are not entirely clear on what the reviewer means. In general, we have tried our best to tighten the text and avoid redundancies and vague words. The paragraph mentioned by the reviewer is revised (page 1, lines 11-13).

  • Major drawback is the lack of supporting references and incomplete experimental and paradigms.

Response: We have incorporated additional references as per reviewer's feedback and have also expanded upon the methodological details in the Methods section. Additionally, we have included more references in the Methods section.

  • A separate detail paragraph required here, to explain the experimental design with detail explanation in flow chart also.

Response: As per reviewer’s suggestion we have updated Fig. 1a (experimental flow chart), depicting the experimental design.

  • How was the sample size determined? Ideally, a priori sample size calculation should be performed to determine the appropriate sample size

Response: Our sample sizes are be similar to previous studies, where we have determined the sample size using G-power analysis for specific behavioral and morphological or molecular experiments (see, for example, Agarwal et al. Neuron 2021, Gangadharan et al. Nature 2022). This is now clarified in the revised methods. As per the animal welfare laws of Germany, we strive to employ the minimum number of mice so as to reduce burden of suffering for animals, while also ensuring adequate numbers for statistical analyses.

  • Normality and variance homogeneity should be assessed across all groups of the same outcome variable and not individual experimental groups. If the data were not normally distributed or variance homogeneity was not met, nonparametric tests need to be performed. Parametric data should be reported as mean +/- sd, while nonparametric data should be given/displayed as median and interquartile range. Longitudinal data should be analyzed using repeated measures tests.

Response: The data in our experiments are normally distributed, based on our own previous experience and the literature. For experiments involving longitudinal analyses, we agree with the reviewer that it is important to analyze with repeated measures tests. We have now performed a 2-way ANOVA for repeated measures on all behavioral data and other such longitudinal data, which pertains to Fig. 1, 2 and 3. These details are added in all figure legends. The error bars represent ‘standard error of the mean’ (S.E.M) and this is uniform across all figures so as to avoid confusion. We have always followed this policy in all of our previous 148 publications, including those on neuropathic pain (Agarwal et al. Neuron 2021, Gangadharan et al. Nature 2022, Gan et al. Science 2023) and never received any contrary opinions.

  • Color expression pattern of all western blot images are not properly arranged. As well as require scale bar and annotation sign in each panel of figure 2a, 2c, 3 c. Without this, manuscript can’t proceed further.

Response: In accordance with the reviewer's suggestions, we have adjusted the Western blots by converting them to grayscale to ensure uniformity. While scale bars were previously present in Figure 2a, 2c, and Figure 3c along with descriptions in the figure legends, we have now added the scale bar text directly onto the figures to enhance clarity.

  • All results are very poorly explained. Revised explanation for all as above mentioned instructions.

Response: While we appreciate the reviewer’s kind interest and advice, this type of comment is not easy to follow since it lacks specifics. We have to respectfully disagree that ‘all results are very poorly explained’.

  • Results need more clarification and significant justification. Differentiating between the outcome and the discussion sections is quite difficult.

Response: We have gone through the text in Results and Discussion sections and worked on differentiating them in the revised manuscript.

  • To address the outcome of results separately, avoiding the disease condition and maintaining physiological condition. How they correlate with the existing literature, it would be better if the author restructured to take a more critical approach of SUMOylation modulates reactive oxygen species (ROS) levels and acts as a protective mechanism in the type 2 model of diabetic peripheral neuropathy.

Response: We regret that we do not understand what exactly the reviewer means. This manuscript is about diabetic neuropathy, which is a pathological condition. We are not addressing physiology in this study. We have generated hypotheses and questioned them stringently using rigorous and quantitative analyses. Please also note that we have recently shown and published that SUMOylation in peripheral nerves represents a protective mechanism against neuropathy in type 1 diabetes (Agarwal et al. Neuron 2023).

  • In the discussion and the conclusion, the aims, rationale, and future perspectives are not evident clearly in relation to in-vitro experimentation.

Response: In response to the reviewer's suggestions, we have incorporated additional details in the Discussion section.

  • The discussion is usually unorganized at the beginning to address and evaluate all the observations at the end. It makes the results easier to contextualize and more straightforward to comprehend.

Response: We are sorry, but the reviewer’s opinion is again not becoming clear in terms of what is meant. We have already written and organized the discussion in a logical order. In our view, it is not necessary to repeat the observations at the beginning of the discussion section, since that just makes it redundant with results and as such, is no longer recommended by scientific journals. Our laboratory has a long record of publications, both numerous in number and high-ranking in quality, and we have a very long-standing experience in writing papers.

  • Furthermore, a minimal critical analysis should be provided, along with current study limitations and the future perspective as separate paragraphs.

Response:  As per reviewer's suggestions, we have incorporated additional details in the Discussion section, which outline the limitations of this study (page 15, lines 546-549). Future perspectives are provided in a short new paragraph (page 15, lines 537-546).

  • Need to revise the conclusion scientifically. Not accepted in its current form.

Response: It is not clear to the authors what is wrong with the conclusions, since no specifics were provided. We have revised the conclusion to the best of our abilities and do stand by the science. Our conclusions are backed by our results.

  • A detailed revision shortening, ordering and following the commented ideas could improve this paper. Several typewriting mistakes are present and need correction. This reviewer remains at the entire disposal for the next version.

Response: We have acted upon the reviewer’s detailed recommendations. We have now proofread the manuscript using the ‘spelling check’ option to correct any typographical errors.

Reviewer 2

Comments and Suggestions for Authors

  • Non-diabetic control missing to rule out the involvement of aging; therefore aging remains a confounding factor.

Response: We fully appreciate the reviewer’s concern. However, the aim of this study was to investigate the role of SUMOylation in progression of diabetic neuropathy. Therefore, we have compared Ubc9fl/fl control mice to SUMOylation-deficient SNS-Ubc9-/- mice under diabetic condition. Please also note that in a previous study, non-diabetic Ubc9fl/fl control mice and non-diabetic SNS-Ubc9-/- mice were analysed in parallel (Agarwal et al, 2020), and we did not find any age-related behavioural differences between the groups.

  • Loading control missing in Figure 3A. An increase in PKG1 dimer/monomer ratio is probably due to decrease in monomer. What are your thoughts on it? Is PKG1 getting degraded? 

Response: We have added a loading control in Fig. 3a. Our thought behind studying PKG1 status in this study was based on published reports that PKG1 serves as a redox sensor through its cysteine oxidation, leading to dimerization and subsequent alterations in the dimer-to-monomer ratio (Burgoyne et al 2007, Valek et al 2017). This process occurs independently of the total PKG protein expression or degradation. We have previously employed PKG1 dimer-to-monomer ratio to determine redox status of cells in pathological conditions (Rojas et al 2018).

  • Magnified images required for Figure 3c. 

Response: We were happy to act on the reviewer’s advice and have inserted a magnified image of the highlighted area within the Fig. 3c.

  • Can you please elaborate why you see increase in GSH with HFD, when HFD has been shown to decrease GSH levels. 

Response: We fully agree with reviewer’s viewpoint on this ambiguous nature of relationship between high fat diet (HFD) and GSH. Indeed, most of the studies performed on the kidney, liver or in serum reported reduction in GSH levels in models of diabetes.  On the contrary, it is noteworthy that several investigations focusing on peripheral nerves have reported an augmentation in both levels and activity of GSH in several models of peripheral neuropathy (Costa et al., 2007; Guedes et al., 2009). Furthermore, an increase in GSH activity has also been documented in the skeletal muscle in a model of chronic neuropathic pain (Tan et al., 2008). Thus, where neuropathy is involved, GSH seems to be modulated differently. We were open in our analyses, and observed an increase in GSH in peripheral nerves upon treatment with HFD. This intriguing discrepancy is now discussed in additional text which we inserted in the discussion section (page 14, lines 471-478).

  • Fig 3g, please correct Complex IV instead of VI on the blot. 

Response: Thank you, we have corrected the labelling as per the reviewer’s suggestion.

  • Have a separate graph for each complex in Figure 3h. At the moment, figure appears misleading. The complex 1 levels in SNS-Ubc9-/- mice + HFD looks more than complex V, III, and IV in the same group which is not true. 

Response: We agree with the reviewer’s suggestion and have now separated the graph for all complexes in revised Fig. 3h.

  • HFD group not included in Figure 4d. 

Response: Unfortunately, making viable DRG cultures is limited to young mice. In the group of HFD-treated mice, mice reach an age of 40 weeks by the time that the HFD induces clear manifestations. It is technically not possible to culture DRGs from 40 weeks-old mice. We have tried and observe rapid cell loss.

  • How can we therapeutically target this in DPN?

Response: This is indeed an interesting and challenging point. Our results point to a protective role of SUMOylation against diabetic peripheral neuropathy, both in type 1 (Agarwal et al. 2021) and type 2 diabetes (this study). Because type 2 diabetes is much more frequent that type 1 diabetes, and represents a common disorder, particularly in developed nations, the implications of our findings are manifold and important. From a therapeutic perspective, enhancing the expression or activity SUMOylation pathway enzymes, specifically in peripheral neurons, may help prevent or reverse the course of diabetic neuropathy. To date, there are no drugs known that activate SUMOylation enzymes with a high level of specificity. However, several drugs can modulate SUMOylation, including agents that are known to be beneficial in diabetics, such as Gingko extract. In recent times, new small molecules are emerging (e.g., Krajnak et al, 2018, Supinski et al, 2020), thus rendering it possible to test these in diabetic neuropathy. Alternately, if specificity concerns remain, it will be helpful to employ gene delivery systems specifically targeting peripheral neurons in the dorsal root ganglia, e.g. as described by Chan et al. (2017) to replenish deficient SUMOylation enzymes. Thus, the future holds attractive perspectives for testing the therapeutic potential of modulating SUMOylation in diabetic neuropathy. We have now included this point in the “Discussion” section.

References:

Agarwal N, Taberner FJ, Rangel Rojas D, Moroni M, Omberbasic D, Njoo C, Andrieux A, Gupta P, Bali KK, Herpel E, Faghihi F, Fleming T, Dejean A, Lechner SG, Nawroth PP, Lewin GR, Kuner R. SUMOylation of Enzymes and Ion Channels in Sensory Neurons Protects against Metabolic Dysfunction, Neuropathy, and Sensory Loss in Diabetes. Neuron. 2020 Sep 23;107(6):1141-1159.e7.

Burgoyne JR, Madhani M, Cuello F, Charles RL, Brennan JP, Schröder E, Browning DD, Eaton P. Cysteine redox sensor in PKGIa enables oxidant-induced activation. Science. 2007 Sep 7;317(5843):1393-7.

Chan KY, Jang MJ, Yoo BB, Greenbaum A, Ravi N, Wu WL, Sánchez-Guardado L, Lois C, Mazmanian SK, Deverman BE, Gradinaru V. Engineered AAVs for efficient noninvasive gene delivery to the central and peripheral nervous systems. Nat Neurosci. 2017 Aug;20(8):1172-1179.

Costa B, Trovato AE, Comelli F, Giagnoni G, Colleoni M. The non-psychoactive cannabis constituent cannabidiol is an orally effective therapeutic agent in rat chronic inflammatory and neuropathic pain.   Eur J Pharmacol. 2007 Feb 5;556(1-3):75-83.

Gan Z, Gangadharan V, Liu S, Körber C, Tan LL, Li H, Oswald MJ, Kang J, Martin-Cortecero J, Männich D, Groh A, Kuner T, Wieland S, Kuner R. Layer-specific pain relief pathways originating from primary motor cortex.  Science. 2022 Dec 23;378(6626):1336-1343.

Gangadharan V, Zheng H, Taberner FJ, Landry J, Nees TA, Pistolic J, Agarwal N, Männich D, Benes V, Helmstaedter M, Ommer B, Lechner SG, Kuner T, Kuner R.  Neuropathic pain caused by miswiring and abnormal end organ targeting. Nature. 2022 Jun;606(7912):137-145. doi: 10.1038/s41586-022-04777-z. Epub 2022 May 25.

Guedes RP, Dal Bosco L, Araújo AS, Belló-Klein A, Ribeiro MF, Partata WA. Sciatic nerve transection increases gluthatione antioxidant system activity and neuronal nitric oxide synthase expression in the spinal cord. Brain Res Bull. 2009 Dec 16;80(6):422-7.

Krajnak K, Dahl R. Small molecule SUMOylation activators are novel neuroprotective agents. Bioorg Med Chem Lett. 2018 Feb 1;28(3):405-409.

Rojas DR, Tegeder I, Kuner R, Agarwal N.  Hypoxia-inducible factor 1α protects peripheral sensory neurons from diabetic peripheral neuropathy by suppressing accumulation of reactive oxygen species.J Mol Med (Berl). 2018 Dec;96(12):1395-1405.

Supinski GS, Wang L, Schroder EA, Callahan LAP. MitoTEMPOL, a mitochondrial targeted antioxidant, prevents sepsis-induced diaphragm dysfunction. Am J Physiol Lung Cell Mol Physiol. 2020 Aug 1;319(2):L228-L238.

Tan EC, Bahrami S, Kozlov AV, Kurvers HA, Ter Laak HJ, Nohl H, Redl H, Goris RJ. The oxidative response in the chronic constriction injury model of neuropathic pain. J Surg Res. 2009 Mar;152(1):84-8.

Valek L, Häussler A, Dröse S, Eaton P, Schröder K, Tegeder I. Redox-guided axonal regrowth requires cyclic GMP dependent protein kinase 1: Implication for neuropathic pain. Redox Biol. 2017 Apr;11:176-191

Reviewer 2 Report

1) Non-diabetic control missing to rule out the involvement of aging; therefore aging remains a confounding factor. 

2)Loading control missing in Figure 3A. An increase in PKG1 dimer/monomer ratio is probably due to decrease in monomer. What are your thoughts on it? Is PKG1 getting degraded? 

3) Magnified images required for Figure 3C. 

4) Can you please elaborate why you see increase in GSH with HFD, when HFD has been shown to decrease GSH levels. 

5) Fig 3g, please correct Complex IV instead of VI on the blot. 

6) Have a separate graph for each complex in Figure 3h. At the moment, figure appears misleading. The complex 1 levels in SNS-Ubc9-/- mice + HFD looks more than complex V, III, and IV in the same group which is not true. 

7) HFD group not included in Figure 4d. 

8) How can we therapeutically target this in DPN? 

Minor editing required. 

Author Response

(The authors gave the same response as above.)

Round 2

Reviewer 1 Report

Dear author, 

after careful revision, manuscript revised successfully, can be proceed further for publication. 

Reviewer 2 Report

Authors addresses my concerns. 

Minor grammar check required.